# FortiNIDS: Defending Smart City IoT Infrastructures Against Transferable Adversarial Poisoning in Machine Learning-Based Intrusion Detection Systems

**DOI:** 10.3390/s25196056

**Published:** 2025-10-02

**Authors:** Abdulaziz Alajaji

**Affiliations:** Information Systems Department, College of Computer and Information Sciences, King Saud University, Riyadh 11451, Saudi Arabia; aalajaji@ksu.edu.sa

**Keywords:** adversarial machine learning, Network Intrusion Detection Systems (NIDS), data poisoning attacks, transferability, random forest, gradient boosting, adversarial training, Reject on Negative Impact (RONI), smart city security, IoT security

## Abstract

In today’s digital era, cyberattacks are rapidly evolving, rendering traditional security mechanisms increasingly inadequate. The adoption of AI-based Network Intrusion Detection Systems (NIDS) has emerged as a promising solution, due to their ability to detect and respond to malicious activity using machine learning techniques. However, these systems remain vulnerable to adversarial threats, particularly data poisoning attacks, in which attackers manipulate training data to degrade model performance. In this work, we examine tree classifiers, Random Forest and Gradient Boosting, to model black box poisoning attacks. We introduce FortiNIDS, a robust framework that employs a surrogate neural network to generate adversarial perturbations that can transfer between models, leveraging the transferability of adversarial examples. In addition, we investigate defense strategies designed to improve the resilience of NIDS in smart city Internet of Things (IoT) settings. Specifically, we evaluate adversarial training and the Reject on Negative Impact (RONI) technique using the widely adopted CICDDoS2019 dataset. Our findings highlight the effectiveness of targeted defenses in improving detection accuracy and maintaining system reliability under adversarial conditions, thereby contributing to the security and privacy of smart city networks.

## 1. Introduction

Cybersecurity is a central concern in today’s digital world. With the rise of smart cities and the Internet of Things (IoT), billions of interconnected devices continuously generate and exchange data. This connectivity creates new opportunities but also exposes systems to a wide range of cyber threats. The sophistication and frequency of cyberattacks have grown rapidly in recent years, resulting in severe economic and social impacts. In 2024, the average cost of a data breach reached USD 4.88 million, the highest ever recorded, and human error contributed to 88% of security incidents [1]. These figures highlight both the scale of the problem and the urgency of developing stronger protection mechanisms.

Traditional security defenses, such as firewalls and signature-based detection, are increasingly inadequate in the face of evolving threats. Signature-based systems can only recognize attacks that match known patterns, leaving them ineffective against novel or polymorphic attacks [2]. Heuristic analysis and manual reviews by analysts provide additional layers of defense, but they often generate excessive false positives and cannot keep pace with fast-moving threats [3]. As a result, researchers and practitioners have turned to artificial intelligence (AI) and machine learning (ML) to design more adaptive and scalable intrusion detection systems (IDS).

AI-based network intrusion detection systems (NIDS) use machine learning algorithms to analyze large volumes of network traffic in real-time, identify anomalies, and detect previously unseen attack vectors. These systems represent an important step forward because they can continuously learn and adapt as new data becomes available. However, despite these advantages, AI-based NIDS also introduce new vulnerabilities. They are prone to common technical issues such as class imbalance and false positives, but more critically, they are vulnerable to adversarial manipulation.

One of the most pressing threats is data poisoning, a type of adversarial attack in which attackers deliberately manipulate the training data to weaken model performance. Poisoning can take several forms, such as injecting mislabeled samples, altering feature distributions, or introducing carefully crafted noise. These malicious modifications compromise the learning process, leading the model to misclassify malicious traffic as benign or to generate high rates of false alarms. In practice, this means that a poisoned IDS may fail to detect intrusions, leaving critical systems exposed to breaches. Given the reliance on AI-driven models in modern cybersecurity, defending against data poisoning is both urgent and necessary.

In addition to poisoning, adversarial examples also threaten AI-based systems. By making small, carefully designed perturbations to input data, adversaries can cause even well-trained models to misclassify traffic. Such examples exploit the sensitivity of machine learning decision boundaries and are often transferable, meaning that an attack crafted for one model can deceive another, even when the models use different architectures or training methods. This property increases the risk of real-world exploitation. Attackers may not need full access to a target model in order to mount a successful attack.

This paper focuses on strengthening the resilience of AI-based NIDS against these adversarial threats. To do so, we evaluate the vulnerability of common classifiers, including Random Forest (RF), Gradient Boosting (GB), and Neural Networks (NN), under controlled poisoning and adversarial attack scenarios using the CICDDoS2019 dataset. We then examine two defense strategies that aim to improve robustness. The first, Reject on Negative Impact (RONI), filters out training samples that harm model performance, thus reducing the influence of poisoned data. The second, adversarial training, involves retraining the model with adversarial examples so that it learns to withstand perturbations. By comparing these approaches, we aim to provide insights into their effectiveness, limitations, and combined potential. Our study distinguishes itself from prior work in two ways. First, we provide the first simultaneous evaluation of RONI filtering and adversarial training across both classical (RF, GB) and deep (NN) models under consistent experimental conditions. Prior research typically examines these defenses in isolation. Second, we analyze cross-model transferability of attacks, quantifying how poisoned or adversarial examples crafted for one model affect others. Together, these contributions extend the literature on adversarial robustness for NIDS by linking complementary defenses with attack transferability.

The contributions of this paper are as follows:We introduce FortiNIDS, a unified evaluation framework that integrates poisoning and gradient-based adversarial attacks, enabling consistent cross-model benchmarking.We conduct a systematic empirical evaluation of adversarial and poisoning attacks on AI-based NIDS, focusing on their impact on detection performance across multiple classifiers.We provide the first side-by-side comparison of adversarial training and RONI defenses under identical conditions, clarifying their complementary strengths.We study the transferability of adversarial examples and measure model divergence, providing insights into how models behave under attack and how defenses influence their stability.We discuss the practical implications of deploying these defenses in real-world NIDS environments, highlighting trade-offs between robustness, accuracy, and computational cost.

The remainder of this paper is organized as follows. Section 2 reviews related work. Section 3 introduces the necessary background concepts. Section 4 presents the system design and methodology. Section 5 describes the evaluation methodology and datasets. Section 6 reports the experimental results, while Section 7 provides a discussion of the findings. Section 8 outlines limitations and directions for future research. Finally, Section 9 concludes the paper.

## 2. Related Work

Recent studies show that AI-driven NIDS are vulnerable to adversarial and poisoning attacks. Gradient-based methods such as the Fast Gradient Sign Method (FGSM) [4] and Projected Gradient Descent (PGD) [5] have been widely tested, and results indicate large performance drops. In many cases, accuracy falls from above 90% to around 60–70% [6,7,8]. Optimization-based attacks such as the Carlini and Wagner (C&W) method further expose weaknesses, lowering accuracy from over 90% to nearly 55–60% in several IDS applications [9,10].

Backdoor and label-flipping poisoning attacks also present serious risks. Studies demonstrate that injecting only 20–30% malicious or mislabeled data during training can reduce accuracy by more than 30% and significantly increase false negative and false positive rates [11,12,13]. These attacks highlight the fragility of IDS models when adversaries target training pipelines.

Several defenses have been explored. Data sanitization, defensive data augmentation, and anomaly detection have shown partial success, restoring accuracy by 15–20% in poisoned settings [14,15,16,17]. Ensemble approaches that combine RF, GB, or XGBoost models have improved resilience by reducing error rates and mitigating accuracy degradation under adversarial conditions [18,19,20].

Most existing work focuses on individual defenses such as adversarial training or data-level preprocessing, often applied to differentiable models. Fewer studies examine transferability of adversarial examples across models or consider black-box attack settings.

In contrast, our framework FortiNIDS is novel in three respects. First, it jointly evaluates RONI and adversarial training across both ensemble models (RF, GB) and NN offering a unified perspective on complementary defenses that prior work studied separately. Second, it systematically measures cross-model transferability between white-box (FGSM, PGD) and black-box poisoning attacks, an aspect underexplored in IDS literature. Third, it incorporates an analysis of computational overhead, providing practical insights for deployment that go beyond accuracy metrics. Together, these contributions distinguish FortiNIDS from existing defenses and highlight its methodological and empirical novelty.

## 3. Background

This section introduces the key concepts required to understand the methodology. We first define adversarial examples, describe white-box and black-box attack settings, explain poisoning strategies, and finally present defense mechanisms including RONI, adversarial training, and model divergence.

### 3.1. Adversarial Examples

Adversarial examples are inputs intentionally perturbed to mislead a machine learning model into making incorrect predictions [4,21]. For a clean input *x* with true label *y*, an adversarial example $x_{\mathrm{adv}}$ is generated as(1)$$x_{\mathrm{adv}}=x+\delta ,\;∥\delta ∥\leq \epsilon$$
where δ is the perturbation and ϵ bounds its magnitude. Although $x_{\mathrm{adv}}$ remains similar to *x*, it causes the model to misclassify.

### 3.2. White-Box and Black-Box Attacks

In white-box attacks, the adversary has full access to the model, including parameters, gradients, and training data [5]. This knowledge allows them to craft precise adversarial examples. A common gradient-based approach is the FGSM [4], which perturbs the input using the gradient of the loss. PGD [5] extends FGSM by applying iterative perturbations within an ϵ-ball, making it a stronger first-order attack.

In black-box attacks, the adversary has no internal knowledge of the model and can only query outputs such as labels or confidence scores [22]. One important property here is transferability, where adversarial examples generated for one model often succeed against others, even with different architectures or training data [22].

### 3.3. Poisoning Attacks

Poisoning attacks target the training phase by injecting harmful samples into the dataset [23]. Several variants exist. In label flipping, correct labels are deliberately altered, misleading the model during training [12,13]. Backdoor insertion embeds hidden triggers into training inputs, which later cause targeted misclassification at inference [9]. Feature corruption modifies selected features to distort the model’s decision boundary [16]. Other strategies, such as gradient manipulation or direct data injection, disrupt the optimization process by introducing adversarially crafted samples [17,24]. Collectively, these techniques bias the learning process and weaken generalization.

### 3.4. Defensive Strategies

Several strategies have been proposed to counter adversarial and poisoning attacks. RONI evaluates each candidate training sample by testing its effect on a validation set. If the inclusion of a sample $x_{i}$ reduces performance by more than a tolerance threshold δ, the sample is discarded [4,21].Adversarial training improves robustness by retraining models on both clean and adversarial examples [21,25]. By exposing the model to perturbed inputs, the decision boundaries are adjusted to resist manipulation.

## 4. System Design and Methodology

In this section, we present the design of FortiNIDS. The system evaluates and mitigates adversarial risk in NIDS across tree ensembles (RF, GB) and NN. We consider black-box poisoning attacks primarily against ensembles and white-box gradient attacks against NN, while evaluating defenses (RONI, adversarial training) across models to assess complementary strengths.

FortiNIDS is structured into two core components: (1) adversarial robustness evaluation, where the susceptibility of models to different attack strategies is tested, and (2) robustness enhancement, where defense mechanisms are applied to strengthen detection reliability. As illustrated in Figure 1, FortiNIDS considers both white-box and black-box settings. White-box attacks (FGSM, PGD) target NN models; black-box poisoning (label flipping, random feature manipulation) targets RF/GB. Although each defense is most naturally aligned to a specific threat (RONI to poisoning; adversarial training to gradient-based attacks), we also evaluate both defenses across models/attacks to quantify complementary benefits and trade-offs.

White-box attacks (FGSM, PGD) target NN, where gradient access is feasible, while black-box poisoning attacks (label flipping, random feature manipulation) are applied to RF and GB, where attackers manipulate the training data instead of gradients. This mapping of attacks to models forms the basis of our experimental design and ensures that each defense strategy (RONI, adversarial training) is evaluated under its most relevant threat scenario.

### 4.1. Model Training

The first step establishes baseline classifiers capable of distinguishing between benign and malicious traffic. Let the input space be represented by feature vectors $x\in R^{d}$ with labels y∈{0,1}, where 0 denotes benign and 1 denotes malicious traffic.

A model $f_{\theta }\left(x\right)$ is trained to approximate the mapping (Equation (Equation 2)):(2)$$f_{\theta }:R^{d}\to \left\{0,1\right\}$$
with parameters θ learned from training data. These baseline models serve as the foundation for robustness evaluation and defensive training.

### 4.2. Adversarial Example Generation

FortiNIDS evaluates resilience under two categories of attacks:White-box attacks: the adversary has complete access to the model parameters and gradients. Two well-known gradient-based methods are applied:
−FGSM, defined in Equation (Equation 3):(3)$$x_{\mathrm{adv}}=x+\epsilon \cdot \mathrm{sign}\left(\nabla_{x}J\left(\theta ,x,y\right)\right)$$
where J(θ,x,y) is the loss function and ϵ controls perturbation magnitude.−PGD: an iterative refinement of FGSM that applies multiple small perturbations and projects adversarial examples back into the valid input space.
Black-box attacks: the adversary lacks internal knowledge of the model and relies on observable outputs. FortiNIDS considers two realistic strategies:
−Random poisoning: injecting perturbed or mislabeled samples at random into the training set.−Label poisoning: deliberately flipping labels of selected benign or malicious samples to corrupt the learning process.


To evaluate transferability, surrogate NN models are trained to approximate the decision boundaries of non-differentiable models like RF and GB. Adversarial perturbations generated on the surrogate are then tested against the target classifiers.

### 4.3. Robustness Enhancement

To counter these attacks, FortiNIDS integrates two complementary defenses:RONI: this defense checks whether a training sample harms model performance. For a candidate sample $x_{i}$, the model is compared when trained with and without $x_{i}$, and the performance change is measured as Equation (Equation 4):(4)$$\Delta P\left(x_{i}\right)=P_{\mathrm{with}\;x_{i}}-P_{\mathrm{without}\;x_{i}}$$If the change is below a negative threshold, i.e., Equation (Equation 5), then $x_{i}$ is considered harmful and is removed:(5)$$\Delta P(x_{i})<-\delta$$This means RONI filters out data points that reduce the effectiveness of the model. By discarding such samples, the final model becomes more resilient to poisoned or misleading data. Figure 2 illustrates the hierarchical workflow of the RONI defense mechanism.Adversarial Training: the model is retrained on a mixture of clean and adversarial examples.In each training round, adversarial samples are generated from the current model using attack algorithms such as FGSM or PGD. These adversarial examples are then combined with the clean dataset to form an augmented training set. The model learns to classify both normal and perturbed inputs correctly, which improves its robustness by adjusting the decision boundaries to resist manipulation. Over multiple iterations, this process strengthens the model against the same types of adversarial perturbations used during training, making it less likely to misclassify maliciously crafted inputs in practice.

## 5. Evaluation Methodology

This section presents the datasets, preprocessing steps, tools, and experimental procedures used to evaluate the proposed system, FortiNIDS. The focus is on how adversarial and poisoning attacks were applied, and how defenses were tested.

### 5.1. Research Aim

The goal is to evaluate the contribution of each defense method, Adversarial Training and RONI, both separately and in combination. This stepwise evaluation makes it possible to identify the unique impact of each technique on system robustness.

### 5.2. Materials

Hardware

All experiments were conducted on a Windows laptop with an Intel^®^ (Core i5 processor, Intel Corporation, Santa Clara, CA, USA) i5 processor and 16 GB of RAM. This configuration was sufficient for training models and running adversarial experiments within a reasonable time.

Software

The experiments were implemented in Python 3.11 [26]. The following libraries and tools were used:Scikit-learn 1.3 for RF, GB, and evaluation metrics [27].Pandas 2.0 and NumPy 1.26 for data manipulation and preprocessing [28,29].We use TensorFlow 2.15 and Keras 2.14 to build surrogate NN models for adversarial attacks [30,31].Matplotlib 3.8 for visualization of results [32].Adversarial Robustness Toolbox (ART) 1.16 and CleverHans 4.0 for implementing adversarial attacks such as FGSM, PGD, and adversarial training [33,34].

### 5.3. Datasets and Preprocessing

The Canadian Institute for Cybersecurity Distributed Denial of Service 2019 (CICDDoS2019) dataset [35] was used in this study. It contains more than 50 million network flow records, including benign traffic and 13 classes of distributed denial-of-service (DDoS) attacks.

For computational manageability, we sampled 12,000 benign records and 25,000 attack records, resulting in a dataset of 37,000 samples labeled as Benignor Malicious.

Preprocessing included converting categorical and non-numeric features into numeric format, imputing missing values using the mean strategy, and standardizing features with z-score normalization. The data were then split into an 80:20 train–test partition, as illustrated in Figure 3, and all attack types were grouped under a single Malicious label versus Benign.

For the NN experiments under adversarial conditions, the dataset was further balanced to address class imbalance. This produced a training set of 29,600 samples and a test set of 7400 samples with 21 features, summarized in Table 1.

### 5.4. Performance Metrics

FortiNIDS is evaluated using both conventional and robustness-oriented metrics:Accuracy, precision, recall, and F1-score measure detection under clean and adversarial conditions.False Positive Rate (FPR) is defined as(6)$$\mathrm{FPR}=\frac{\mathrm{FP}}{\mathrm{FP}+\mathrm{TN}}.$$Model divergence quantifies the proportion of test samples that are misclassified by two models simultaneously. Formally, divergence between two models $M_{1}$ and $M_{2}$ is given by(7)$$D\left(M_{1},M_{2}\right)=\frac{1}{N}\sum_{i=1}^{N}I_{i},$$
where the indicator function $I_{i}$ is defined as(8)$$I_{i}=\left\{\begin{array}{cc} 1, & \mathrm{if}\;M_{1}\left(x_{i}\right)\neq y_{i}\;\;\mathrm{and}\;\;M_{2}\left(x_{i}\right)\neq y_{i},\\ 0, & \mathrm{otherwise}. \end{array}\right.$$
where $x_{i}$ is the *i*-th input sample, $y_{i}$ its true label, and *N* the total number of test samples. Equations (Equation 7) and (Equation 8) together capture whether both models fail on the same instance. A higher divergence indicates shared vulnerabilities (greater transferability of adversarial examples), while lower divergence suggests complementary robustness.Transfer success (TS) measures the ability of adversarial or poisoned samples crafted on a source model $M_{s}$ to induce misclassification on a target model $M_{t}$. It is defined as(9)$$\mathrm{TS}\left(M_{s}\;\to \;M_{t}\right)\;=\;\frac{1}{N}\sum_{i=1}^{N}I\;\left[M_{t}\;\left(a(M_{s},x_{i})\right)\neq y_{i}\right],$$
where $a(M_{s},x_{i})$ denotes the adversarial or poisoned transformation of input $x_{i}$ derived from $M_{s}$, and I[·] is the indicator function. Equation (Equation 9) explicitly captures cross-model attack transferability: higher values indicate that attacks created on $M_{s}$ are more successful when applied to $M_{t}$.

### 5.5. Experimental Procedures

#### 5.5.1. White-Box Attacks

Two gradient-based methods were implemented using ART.

PGD

PGD was configured with the $\ell_{\infty }$ norm, ϵ=0.1, step size α=0.01, and T=10 iterations (see Algorithm 1). These values follow common configurations reported in adversarial machine learning literature [4,5]. They were fixed across experiments to ensure comparability rather than to maximize attack success.    
**Algorithm 1:** PGD for adversarial evaluation with T=10, ϵ=0.1, α=0.01**Input**: Classifier $f_{\theta }$, dataset $X_{\mathrm{test}},y_{\mathrm{test}}$**Output**: Evaluation metrics on adversarial samplesStep 1: Load dataset and train surrogate NN.Step 2: Configure PGD attack parameters.Step 3: Generate adversarial examples $X_{\mathrm{adv}}$.Step 4: Evaluate predictions on $X_{\mathrm{adv}}$.Step 5: Report accuracy, precision, recall, and F1-score.

FGSM

FGSM was implemented with $\ell_{\infty }$ constraints, ϵ=0.1, and step size α=0.01. These parameters are widely used in prior work [4] and were kept consistent with PGD settings for fairness (see Algorithm 2).    
**Algorithm 2:** FGSM-based adversarial evaluation**Input**: Classifier $f_{\theta }$, dataset $X_{\mathrm{test}},y_{\mathrm{test}}$**Output**: Evaluation metrics on adversarial samplesStep 1: Train NN model.Step 2: Configure FGSM parameters.Step 3: Generate adversarial inputs $X_{\mathrm{adv}}$.Step 4: Evaluate model performance on $X_{\mathrm{adv}}$.Step 5: Report accuracy, precision, recall, and F1-score.

#### 5.5.2. Black-Box Poisoning Attacks

These attacks are applied at training time to tree ensembles (RF, GB) by modifying inputs and/or labels rather than gradients. The poisoning rates used in our experiments are intentionally strong: 50% random perturbation with Gaussian noise and feature shifts, with up to 90% label flips. While such levels may exceed what is typically seen in practice, they provide a worst-case stress test that highlights the resilience of defenses under extreme conditions. We confirmed that models remained trainable and evaluable at these rates, ensuring the results remain meaningful.

Random Poisoning

Fifty percent of training samples were perturbed with Gaussian noise, feature shifts, and up to 90% label flips (see Algorithm 3). This poisoning intensity reflects a strong but manageable adversary, while still allowing the models to be retrained and evaluated reliably.    
**Algorithm 3:** Random poisoning attack evaluation**Input**: Training dataset $\left(X_{\mathrm{train}},y_{\mathrm{train}}\right)$, test dataset $\left(X_{\mathrm{test}},y_{\mathrm{test}}\right)$**Output**: Comparison of clean vs. poisoned modelsStep 1: Train clean baseline model.Step 2: Poison 50% of training samples with noise, feature shifts, and label flips.Step 3: Retrain model on poisoned dataset.Step 4: Compare accuracy, F1-score, and confusion matrix against baseline.

Label Poisoning

Thirty percent of benign samples were relabeled as malicious based on model confidence (see Algorithm 4). This mid-range poisoning rate simulates systematic mislabeling without overwhelming the dataset.    
**Algorithm 4:** Label poisoning attack evaluation**Input**: Dataset $\left(X_{\mathrm{train}},y_{\mathrm{train}}\right)$, poisoning rate 30%**Output**: Performance comparison and feature importance shiftsStep 1: Select benign samples from dataset.Step 2: Flip labels of 30% of selected samples.Step 3: Retrain clean model and poisoned model separately.Step 4: Compare accuracy, F1-score, and feature importance between models.

#### 5.5.3. Robustness Enhancement

RONI

In our experiments, RONI was applied with a tolerance threshold δ set to 1%. This value was chosen as a balance: small enough to identify harmful data, but not so strict as to reject benign variations [4,21].

Adversarial Training

Models were retrained over three rounds on a mixture of clean and adversarial samples generated with FGSM and PGD. This iterative process progressively adjusts classifier decision boundaries to resist adversarial perturbations. The decision to use three rounds was based on preliminary tuning runs that compared between 1 and 5 rounds of adversarial retraining. As shown in Figure 4, performance improvements saturated after the third round, with accuracy and F1-score gains beyond this point being marginal (below 0.3%). Additional rounds also introduced higher computational overhead and risk of overfitting to adversarial samples. Three rounds therefore represented a principled compromise: sufficient to reinforce model robustness against both FGSM and PGD, while ensuring training efficiency and consistency across RF, GB, and NN models. For ensembles (RF/GB) under poisoning, adversarial training is used as a robustness heuristic against feature-space shifts despite the absence of gradients; for NNs, it serves its conventional role against gradient-based attacks.

## 6. Results

### 6.1. Baseline Training Results

The comparative analysis in Figure 5 presents the classification performance of RF and GB models on the CICDDoS2019 dataset. Both models achieved high classification accuracy above 99% across metrics, though RF consistently outperformed GB in terms of precision, recall, and F1-score for attack traffic. RF reached nearly perfect scores, highlighting its robustness in identifying malicious traffic, whereas GB showed a slight trade-off between recall and precision, leading to a modest increase in false positives.

### 6.2. Adversarial Machine Learning Attacks

#### 6.2.1. White-Box Attacks (NN)

NN models were evaluated under white-box adversarial attacks. On clean data, the NN achieved strong accuracy (98.40%) and F1-score (98.50%). However, adversarial examples degraded performance significantly: FGSM reduced accuracy to 85.70% and PGD further to 79.20%. PGD caused the sharpest reduction across precision, recall, and F1, confirming its greater effectiveness at destabilizing NN predictions (Figure 6).

#### 6.2.2. Black-Box Attacks (RF and GB)

Random Forest

Both label poisoning and random poisoning reduced RF robustness. Label poisoning caused a modest F1 decline (from 97.20% to 95.70%), whereas random poisoning led to a larger accuracy drop (from 96.00% to 91.70%) with sharper reductions in precision and recall. These degradations are summarized alongside defenses; the standalone attack-only plot is omitted for brevity.

Gradient Boosting

GB also degraded under both attacks. Label poisoning reduced accuracy slightly (from 97.80% to 96.60%), while random poisoning caused a larger decline (to 91.70%). A consolidated view with defenses appears in Figure 7 (random poisoning) and Figure 8 (label poisoning); the attack-only plot is omitted for brevity.

### 6.3. Defenses and Mitigation Approaches

#### 6.3.1. RONI Filtering

RONI defenses were first applied against poisoning attacks on RF and GB. Figure 9 and Figure 10 show a three-way comparison of Clean, Poisoned, and Defended performance. For RF (Figure 9), label poisoning degraded accuracy from 99.47% to 91.83% and increased FPR above 20%. RONI filtering restored accuracy to 96.78% and reduced FPR below 7%. Similarly, random poisoning caused accuracy to fall to 90.59%, but RONI defense recovered it to 95.88% with FPR near 3.6%.

For GB (Figure 10), label poisoning reduced accuracy to 87.42% and F1 to 91.51%, while random poisoning lowered accuracy to 95.50%. RONI defense significantly restored performance in both cases, recovering accuracy above 94.6% for label poisoning and nearly 96% for random poisoning, with sharp reductions in FPR.

For NN, RONI was applied as a training-time filter on candidate samples, including both clean and adversarially augmented data. The purpose was to discard training points whose inclusion degraded validation performance beyond the threshold δ, rather than applying RONI at inference time. NN models were also evaluated with RONI filtering under FGSM and PGD. Figure 11 and Figure 12 show that while adversarial samples degraded accuracy sharply (to 83–86%), RONI filtering restored 2–3% of lost accuracy and reduced FPR to nearly zero, maintaining precision close to 100%.

#### 6.3.2. Adversarial Training

Adversarial training was next applied against poisoning attacks. Figure 13 and Figure 14 show clean, poisoned, and adversarially trained models.

For RF (Figure 13), label poisoning increased FPR drastically to over 70%. Adversarial training did not restore robustness in this case, highlighting its ineffectiveness under label corruption. Against random poisoning, however, adversarial training maintained high accuracy (∼99.55%), stabilizing RF performance.

For GB (Figure 14), adversarial training was more effective. Under label poisoning, accuracy was restored from 87.7% to 96.4%, and under random poisoning from 91.7% to 96.5%. This demonstrates that adversarial training can substantially recover GB robustness when labels remain partially reliable.

Training convergence and performance evolution (Figure 15) show stable improvements, with post-training accuracy exceeding 94–95% under attacks, significantly higher than the undefended case (∼83–90%). NN adversarial training was then tested against FGSM and PGD (Figure 16 and Figure 17).

#### 6.3.3. Comparison of Defenses

Finally, Figure 7 and Figure 8 compare RONI and adversarial training directly.

For RF under label poisoning (Figure 8), RONI filtering outperformed adversarial training, reducing FPR from 71.8% to under 7%. Conversely, under random poisoning (Figure 7), adversarial training preserved near-clean accuracy, whereas RONI provided only partial recovery. Thus, RONI is superior for label integrity attacks, while adversarial training is preferable for random feature corruption.

Beyond classification performance, we also measured the computational cost of each defense. Since training overhead is critical for deployment feasibility, a detailed analysis of runtime implications is presented in Section 7.

### 6.4. Model Divergence Metrics

Transferability success and model divergence were further analyzed to evaluate cross-model attack success (Table 2, Table 3 and Table 4). Results confirm that poisoning-induced errors can transfer effectively between clean, poisoned, and adversarially trained models, though RONI filtering consistently reduced divergence.

## 7. Discussion

Our results demonstrate that both RONI-based filtering and adversarial training can improve resilience against adversarial machine learning attacks, though their effectiveness depends strongly on the attack type and the underlying model.

### 7.1. Interpretation of Defense Effectiveness

RONI filtering was consistently effective against poisoning attacks where label integrity was compromised. By rejecting outlier samples that excessively degrade performance, RONI reduced false positive rates from over 70% to below 7% in RF, and from more than 30% to under 8% in GB. These improvements suggest that RONI is well-suited for scenarios where data quality is compromised at the labeling stage. In contrast, adversarial training offered little benefit in these cases because retraining on corrupted labels reinforces incorrect associations rather than correcting them. This explains the unexpected failure of adversarial training under label poisoning observed in our experiments.

For random poisoning attacks, however, adversarial training was more effective than RONI. Since these attacks modify feature distributions without consistently corrupting labels, adversarial retraining helped the model adapt to perturbed inputs and maintain high accuracy (above 99%). In contrast, RONI filtering could only partially restore performance, as poisoned samples blended into the feature space without being detected as clear outliers. This highlights a fundamental limitation of filtering defenses when adversarial perturbations mimic normal variation.

NN displayed similar trends: RONI filtering provided marginal but consistent gains by suppressing adversarial outliers, while adversarial training yielded substantial improvements under FGSM and PGD. The discrepancy between shallow (RF, GB) and deep (NN) models suggests that defense mechanisms may interact with model complexity, an avenue that merits further study.

### 7.2. Combining RONI and Adversarial Training

Our findings suggest that RONI and adversarial training are complementary rather than competing strategies. RONI excels in filtering corrupted labels before training, whereas adversarial training strengthens robustness against feature-space perturbations. A hybrid defense pipeline could first apply RONI to sanitize the dataset, followed by adversarial retraining to improve resilience against distributional shifts. Such a combination could mitigate the shortcomings of each method: RONI’s weakness in handling subtle feature perturbations, and adversarial training’s vulnerability when labels are corrupted. We propose that future work explicitly explore this integrated defense strategy, evaluating whether it can generalize across attack types and model classes.

### 7.3. Computational Overhead Analysis

Figure 18 shows that RONI adds ∼4.2 min (+18%) over baseline, adversarial training increases time to ∼49.1 min (≈2.10×), and the combined configuration totals ∼59.0 min (≈2.52×).

### 7.4. Implications

The results reveal key trade-offs for practitioners. Ensemble methods such as RF and GB are highly robust under clean training conditions, but their vulnerability diverges depending on the poisoning method. Label integrity is critical for maintaining ensemble reliability, whereas feature-level noise is more effectively countered by adversarial retraining. For NN, adversarial robustness requires explicit training, as filtering alone cannot fully address gradient-based white-box attacks.

## 8. Future Work and Limitations

This research has several limitations that warrant discussion. On the practical side, the deployment of RONI and adversarial training in real systems must consider computational overhead, false rejection of legitimate samples, and the dynamic nature of adaptive adversaries. Defenses that succeed in static settings may require ongoing adaptation to remain effective in adversarial environments.

From a methodological standpoint, our experiments were limited to single-label classification, whereas many practical intrusion detection systems require multi-label outputs to model overlapping or correlated attacks. Extending adversarial generation techniques to multi-label settings remains an important future direction. Second, we considered only feature-wise bounded perturbations (e.g., with ϵ=0.1 in the normalized feature space) to restrict adversarial modifications. While practical, this approach does not account for more sophisticated perturbations such as semantic or temporal manipulations, which are increasingly relevant in IoT environments. Third, we did not explore data augmentation techniques, which may enhance model generalization and robustness, particularly in scenarios with imbalanced datasets or limited labeled samples.

Finally, our evaluation relied solely on the CICDDoS2019 dataset. Although widely used, this dataset may not fully capture the complexity and scale of real-world IoT traffic. Future work should therefore include experiments on larger, more diverse IoT datasets and a broader range of attack types to confirm the robustness and scalability of both attack and defense mechanisms in FortiNIDS.

## 9. Conclusions

This study empirically evaluated the robustness of machine learning-based Network Intrusion Detection Systems under adversarial and poisoning attacks. Using the CICDDoS2019 dataset, we demonstrated that RF, GB, and NN classifiers are highly susceptible to both gradient-based adversarial examples and data poisoning strategies. When unprotected, these models experienced significant performance degradation across accuracy, precision, recall, and F1-score.

To mitigate these threats, we examined two defense mechanisms: RONI and adversarial training. RONI effectively filtered suspicious training data, especially against label poisoning, while adversarial training enhanced the resilience of NN against gradient-based perturbations. However, neither defense alone provided complete protection, indicating the necessity of combining multiple strategies for practical deployment.

In future work, we plan to extend the evaluation to multi-label intrusion detection, incorporate semantic and temporal perturbations, and validate robustness on more diverse IoT datasets. Our findings suggest the importance of layered defense approaches in securing AI-based NIDS for real-world applications.

## Figures and Tables

**Figure 1 sensors-25-06056-f001:**
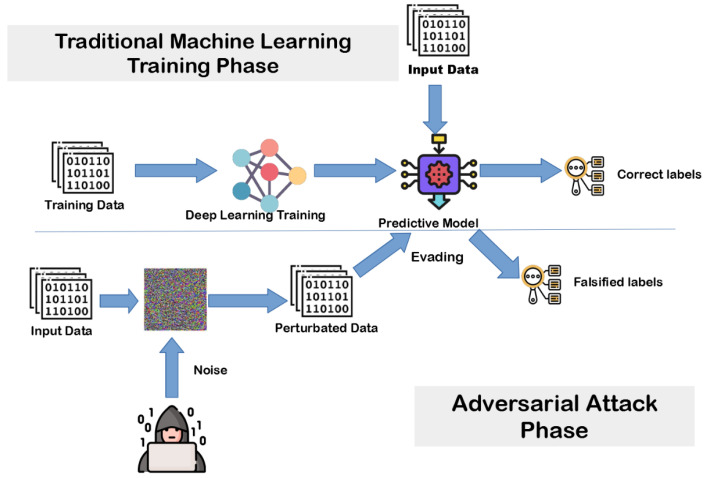
System design of FortiNIDS showing attack–model mapping. White-box gradient attacks (FGSM, PGD) apply to NN, while black-box poisoning attacks target RF and GB. Defenses (RONI, adversarial training) are integrated in the training cycle to mitigate these threats.

**Figure 2 sensors-25-06056-f002:**
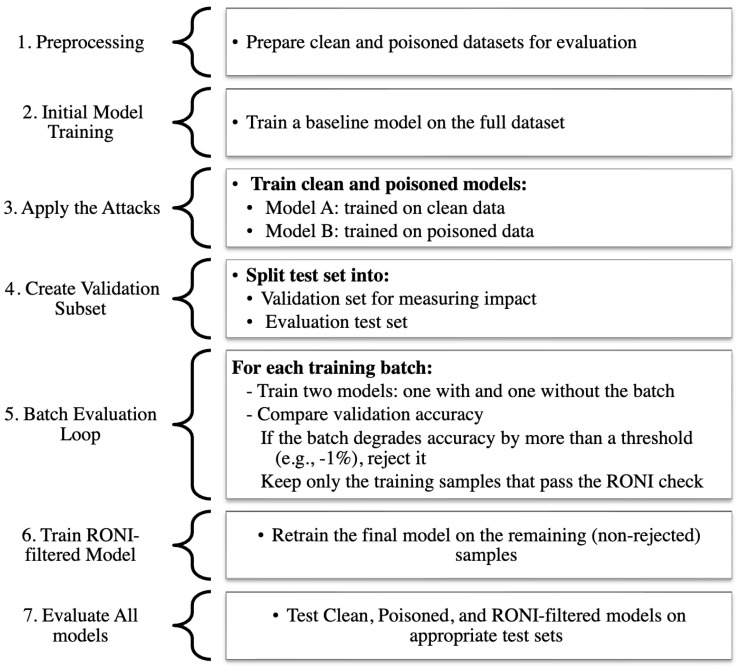
Hierarchical representation of the RONI defense mechanism.

**Figure 3 sensors-25-06056-f003:**
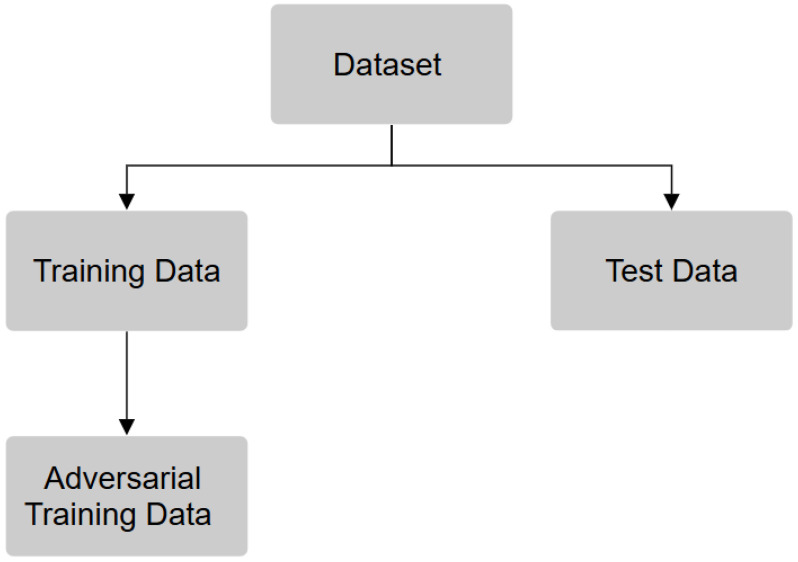
Training and testing data partitioning for the IDS model.

**Figure 4 sensors-25-06056-f004:**
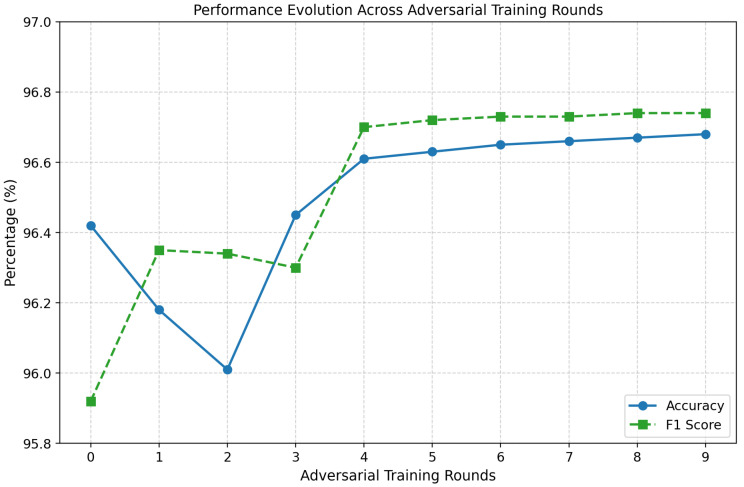
Performance evolution across adversarial training rounds (0–4). Accuracy and F1-score improvements plateau after the third round, supporting the choice of three rounds for consistent and efficient robustness gains.

**Figure 5 sensors-25-06056-f005:**
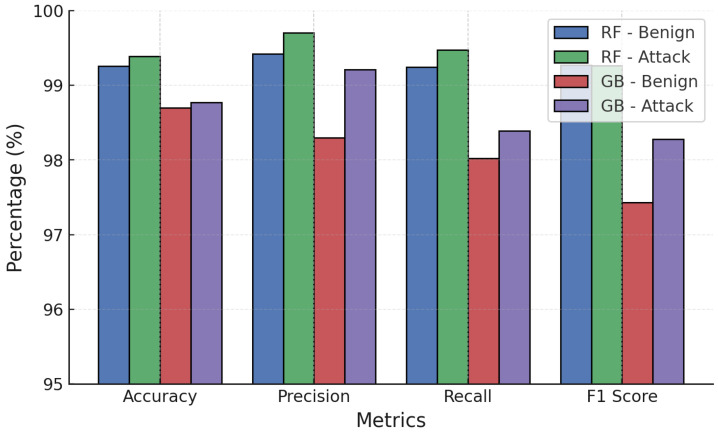
RF vs. GB baseline performance on CICDDoS2019 dataset.

**Figure 6 sensors-25-06056-f006:**
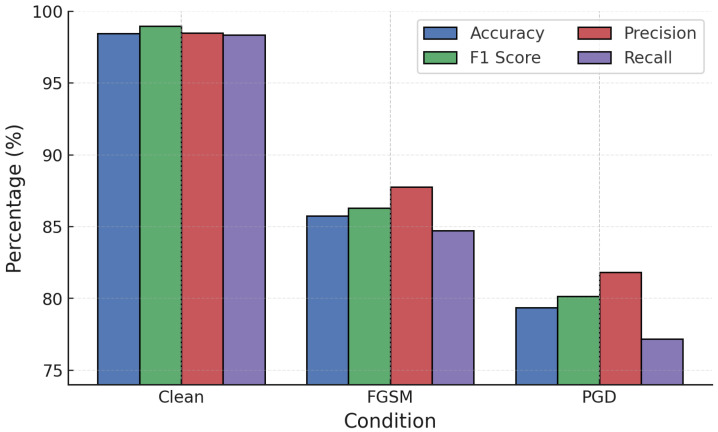
NN performance under FGSM and PGD white-box attacks.

**Figure 7 sensors-25-06056-f007:**
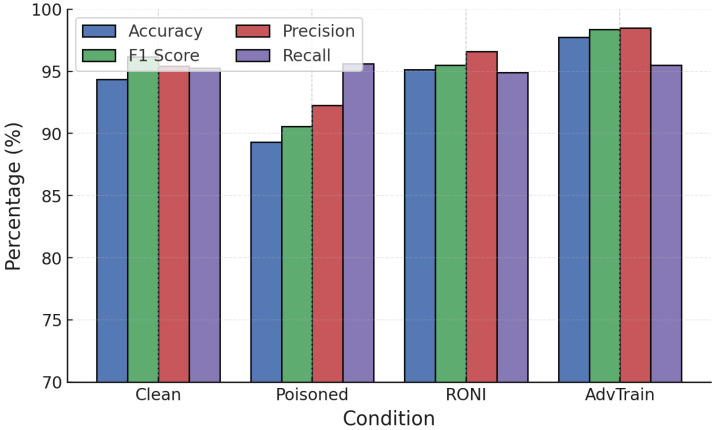
Comparison of RONI vs. adversarial training under random poisoning (RF, GB).

**Figure 8 sensors-25-06056-f008:**
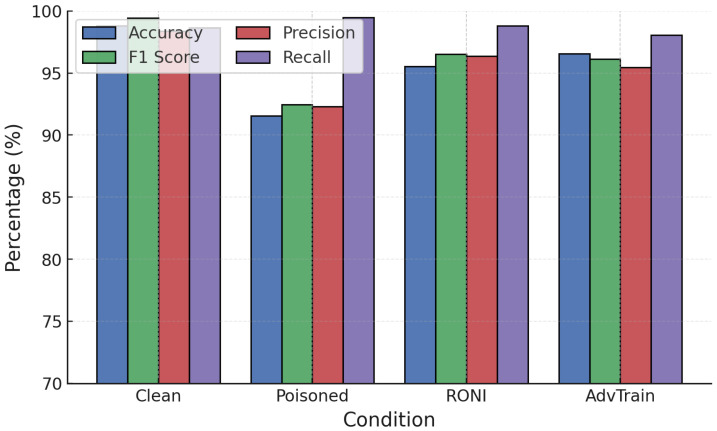
Comparison of RONI vs. adversarial training under label poisoning (RF, GB).

**Figure 9 sensors-25-06056-f009:**
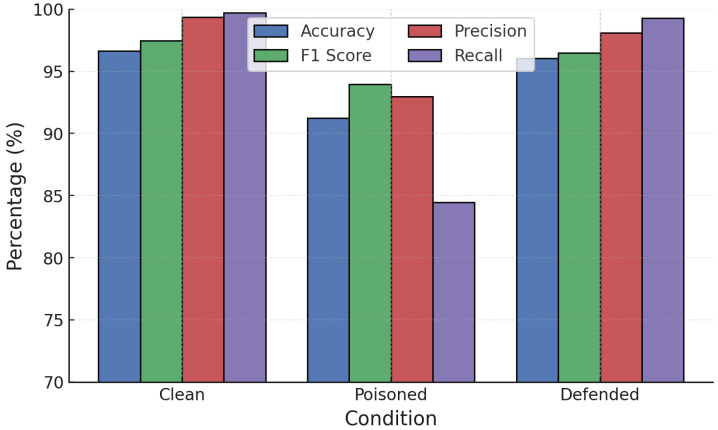
RF performance under poisoning attacks with RONI defense.

**Figure 10 sensors-25-06056-f010:**
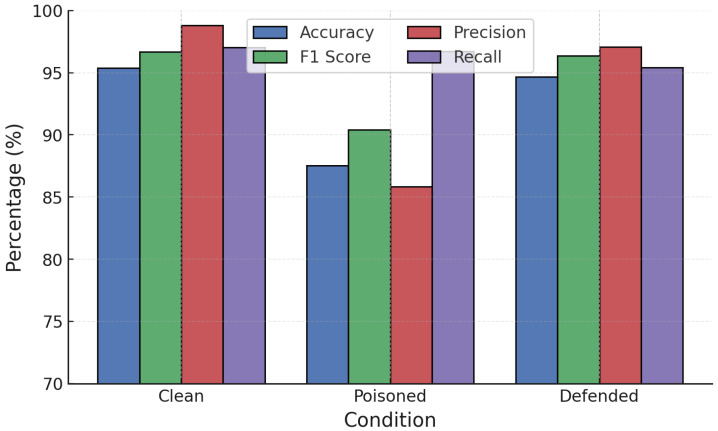
GB performance under poisoning attacks with RONI defense.

**Figure 11 sensors-25-06056-f011:**
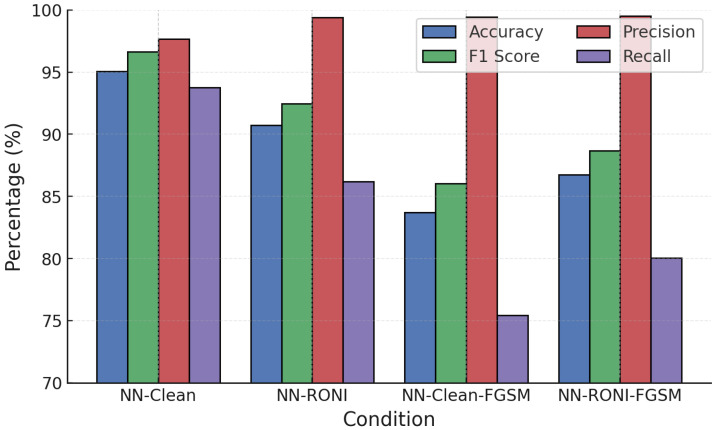
NN performance under FGSM attack with RONI defense.

**Figure 12 sensors-25-06056-f012:**
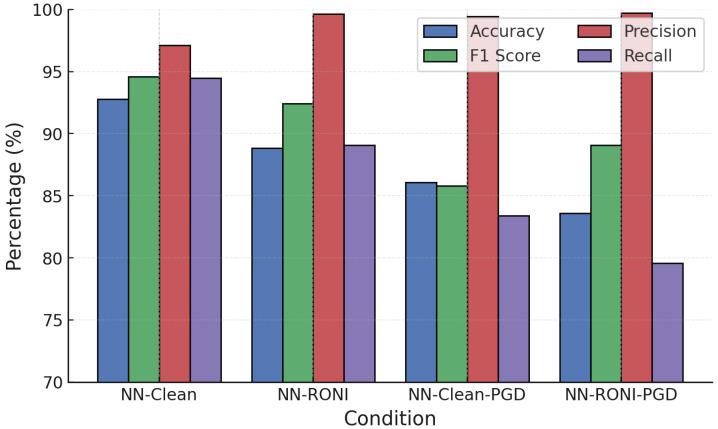
NN performance under PGD attack with RONI defense.

**Figure 13 sensors-25-06056-f013:**
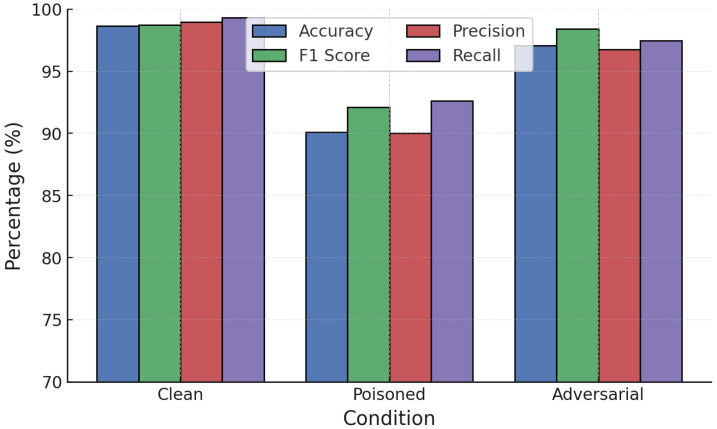
RF performance under poisoning attacks with adversarial training.

**Figure 14 sensors-25-06056-f014:**
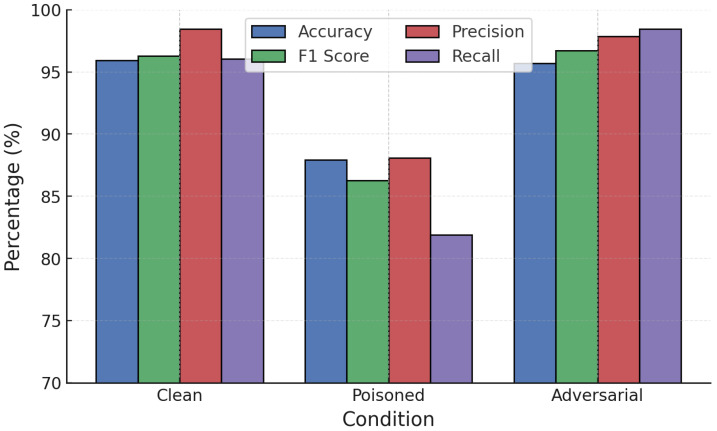
GB performance under poisoning attacks with adversarial training.

**Figure 15 sensors-25-06056-f015:**
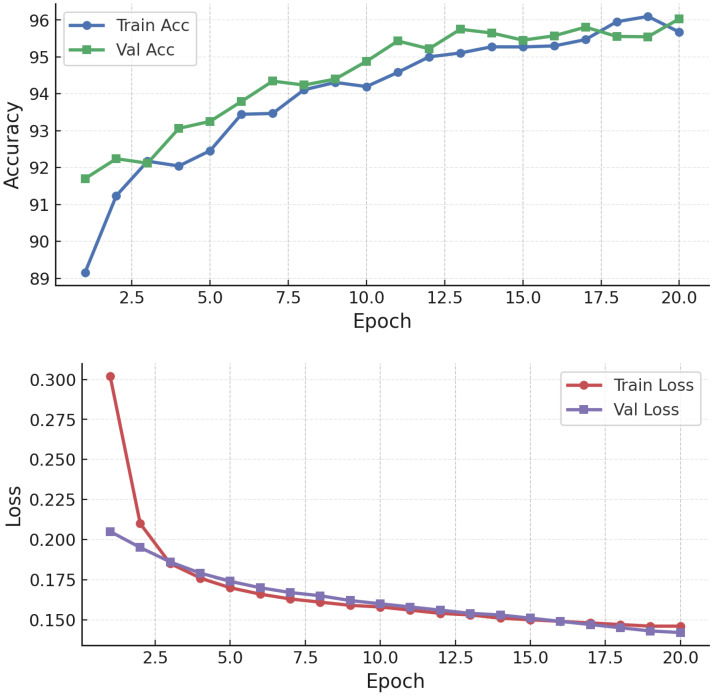
NN training dynamics under PGD adversarial training.

**Figure 16 sensors-25-06056-f016:**
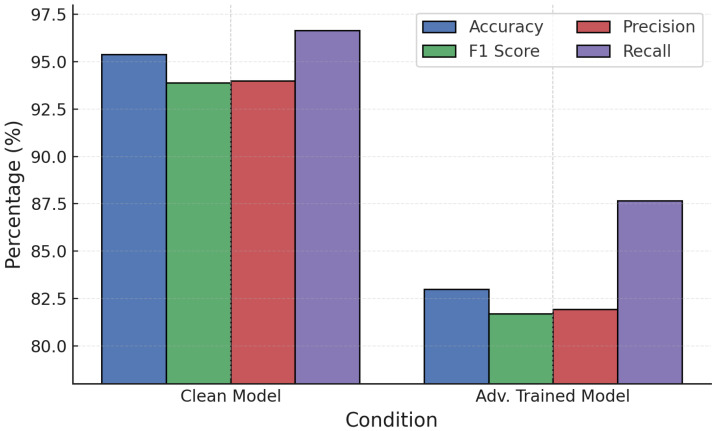
NN performance under PGD attack with adversarial training.

**Figure 17 sensors-25-06056-f017:**
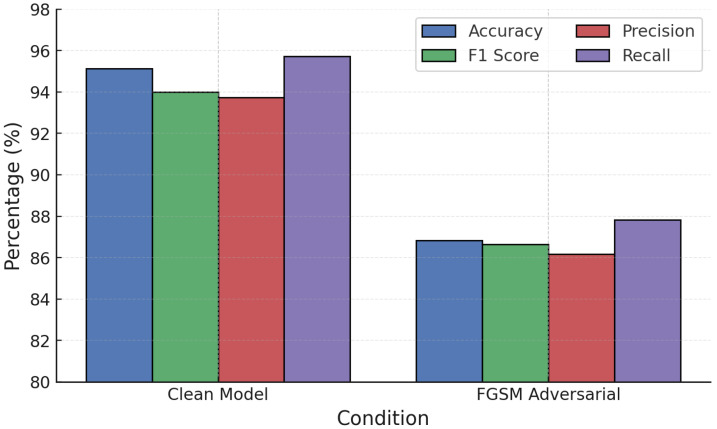
NN performance under FGSM attack with adversarial training.

**Figure 18 sensors-25-06056-f018:**
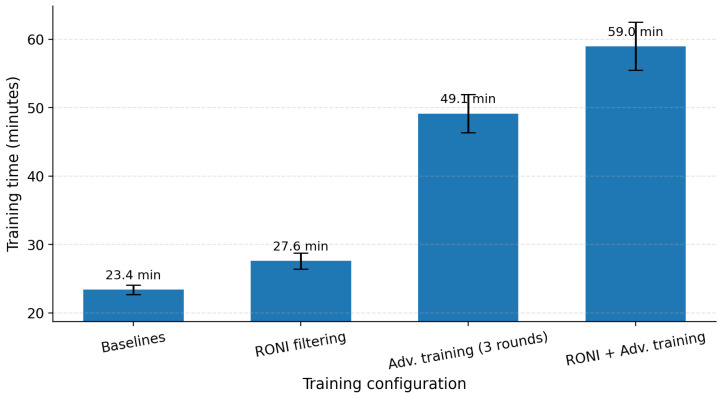
Training time (mean ± SD, minutes) on CICDDoS2019. Baseline = 23.4 ± 0.7; RONI = 27.6 ± 1.2 (1.18×); adversarial training (three rounds) = 49.1 ± 2.8 (2.10×); RONI + adversarial training = 59.0 ± 3.5 (2.52×).

**Table 1 sensors-25-06056-t001:** Balanced dataset splitting and feature information. The symbol “#” denotes the number of instances or features.

Training Set #	Test Set #	Features #
29,600	7400	21

**Table 2 sensors-25-06056-t002:** Transferability success and model divergence for GB.

Model	Attack	Defense	ML Model Pair	Transfer Success / Divergence
GB	Random Poisoning	Adv Training	Clean → Poisoned	99.55% / 0.45%
GB	Random Poisoning	Adv Training	Clean → Adversarial	95.85% / 4.15%
GB	Random Poisoning	Adv Training	Poisoned → Adversarial	95.87% / 4.13%
GB	Random Poisoning	RONI	RONI Filtered Model	57.32% / 2.06%
GB	Label Poisoning	Adv Training	Clean → Poisoned	10.28% / 10.28%
GB	Label Poisoning	Adv Training	Clean → Adversarial	99.85% / 2.00%
GB	Label Poisoning	Adv Training	Poisoned → Adversarial	100.00% / 0.00%
GB	Label Poisoning	RONI	RONI Filtered Model	10.28% / 8.57%

**Table 3 sensors-25-06056-t003:** Transferability success and model divergence for RF.

Model	Attack	Defense	ML Model Pair	Transfer Success / Divergence
RF	Random Poisoning	Adv Training	Poisoned → Adversarial	100.00% / 0.00%
RF	Random Poisoning	Adv Training	Clean → Adversarial	99.85% / 2.00%
RF	Random Poisoning	RONI	RONI Filtered Model	6.22% / 3.61%
RF	Random Poisoning	RONI	Clean → Poisoned	46.37% / 4.27%

**Table 4 sensors-25-06056-t004:** Transferability success and model divergence for NN.

Model	Attack	Defense	Transfer Success	Divergence
NN	FGSM	Adv Training	3.73%	13.16%
NN	FGSM	RONI	12.30%	5.14%
NN	PGD	Adv Training	13.32%	10.24%
NN	PGD	RONI	10.64%	4.82%

## Data Availability

The CICDDoS2019 dataset used in this study is publicly available at: https://www.unb.ca/cic/datasets/ddos-2019.html (accessed on 29 September 2025).

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
