# Peer review of "FortiNIDS: Defending Smart City IoT Infrastructures Against Transferable Adversarial Poisoning in Machine Learning-Based Intrusion Detection Systems"

_sensors, 2025, doi:10.3390/s25196056_

Round 1
Reviewer 1 Report
Comments and Suggestions for Authors
This paper addresses the issue of adversarial poisoning attacks in machine learning-based Network Intrusion Detection Systems (NIDS) within smart city IoT environments. It proposes a defense framework named FortiNIDS, which integrates two defense mechanisms: Reject on Negative Impact (RONI) and adversarial training. The research is well-designed, experiments are comprehensive, and the analysis of results is detailed. The study holds considerable theoretical and practical value for enhancing the robustness of NIDS in adversarial settings.
1. Clearly highlight the distinctions between this study and existing work (e.g., simultaneous evaluation of RONI and adversarial training, cross-model transferability) in the Introduction or Related Work section.
2. In the Experiments section, explicitly justify the choice of "3 rounds" for adversarial training (e.g., whether hyperparameter tuning was performed).
3. Further discuss the potential of combining RONI and adversarial training in the Discussion section.
4. Include a brief analysis of computational overhead (e.g., time cost of RONI filtering or adversarial training), which would be valuable for practical deployment.
Author Response
Thank you for your constructive feedback. Please find our responses attached.

Reviewer 2 Report
Comments and Suggestions for Authors
The manuscript entitled "FortiNIDS: Defending Smart City IoT Infrastructures Against Transferable Adversarial Poisoning in Machine Learning-Based Intrusion Detection Systems" aims to introduce FortiNIDS, a robust framework that employs a surrogate Neural Network model to generate transferable adversarial perturbations, leveraging the transferability property of adversarial examples across models.
The paper is well written, easy to follow, and brings contributions to the domain. Some suggestions for the authors are presented below:
Thus manuscript should be improved in the following main directions:
- the connection between the white-box (FGSM, PGD) and black-box (poisoning) attacks and the specific machine learning models (Random Forest, Gradient Boosting, neural networks) is often confusing. The authors should better explain why certain attacks were tested on specific models and integrate the figures and tables more effectively into the main text. For instance, figure 1 is insufficiently described in lines 157-158.
- the paper introduces FortiNIDS as a framework, but the novelty compared to prior work is not enough highlighted. Thus, the contributions should be better distinguished from existing defenses (e.g., adversarial training, RONI) to clarify what is genuinely new in methodology and/or findings.
- results section includes many (I would say too many) tables and plots but lacks deeper interpretation. Maybe grouping them in fewer figures will be less confusing for the reader. Further, why certain defenses succeed or fail across models is only briefly mentioned. A stronger theoretical or empirical analysis of defense trade-offs (accuracy, computational cost, false positives) would add significant value.
- though the paper has a discussion section (Section 7), it is underdeveloped. It is crucial for the authors to interpret the results in depth, provide explanations for unexpected findings (for example the failure of adversarial training. This section should offer context for the results, not merely summarize them.
- to esure methodology replication, the authors must bring in more information about the preprocessing of the CICDDoS2019 dataset by including the balancing technique used and the specific hardware configuration utilized (the exact CPU and GPU models, etc).
Minor issues:
- the authors should be consequent with the use of acronyms. once defined, used them as acronyms. check lines 37, 301,450 for machine learning, which was defined in line 35. similarly with the acronym NIDS. the authors should check the rest of the document for similar issues;
- the equations should be numbered and referred to as such in the entire material.
Author Response

(The authors gave the same response as above.)

Round 2
Reviewer 1 Report
Comments and Suggestions for Authors
The revised manuscript can be accepted.
Reviewer 2 Report
Comments and Suggestions for Authors
The authors have complied with all of my suggestions. As a result, the new version of the manuscript is greatly improved.
Based on the above, I believe the updated version of the manuscript can be considered for being published with Sensors.